# Research on the mechanism of promoting coordinated development of ecological well-being in rural counties through industrial transformation

Fan Yang[1]*, Wanlin Qi[2], Jiaqi Han[3]

**1** Inner Mongolia University of Finance and Economics, Hohhot, Inner Mongolia, China, **2** Beijing Technology and Business University, Beijing, China, **3** Shenyang Normal University, Shenyang, Liaoning, China

* 15648031080@163.com

**Data Availability Statement:** All relevant data are within the manuscript and its Supporting Information files.

**Funding:** The authors received no specific funding for this work.

## Abstract

The balanced development of ecological quality and residents' well-being is an important factor in achieving sustainable economic development in rural counties. In order to promote the improvement of the coupling coordination degree between ecology and well-being in rural counties, this study explores the impact mechanism of industrial structural transformation and upgrading on the coupling of ecology and well-being in the Sichuan-Chongqing. A dual-fixed-space Durbin model is constructed to analyze the influence mechanism and spatial interaction effects of industrial transformation and upgrading on the coordination of ecology and well-being. The research indicates: (1) From 2010 to 2020, the overall coordination degree of ecology and well-being in the Sichuan-Chongqing rural counties has steadily increased, with higher coordination in the eastern region and lower coordination in the western region. The growth rate of coordination degree is faster in the east and slower in the west, with significant and increasing differences between the east and west. 2) The coupling coordination degree of ecology and well-being in the Sichuan-Chongqing rural counties has a strong positive spatial spillover effect. (3) The more reasonable the industrial structure, the higher the level of coordinated development in the local and surrounding rural counties. The higher the index of industrial advancement, the better the level of coordinated development locally, but the lower the level of coordinated development in the surrounding areas.

## 1 Introduction

With the rapid development of the socio-economic landscape, human activities have led to an exponential expansion in the demand for natural resources [1]. This has resulted in a continuous degradation of ecosystem services, directly or indirectly impacting human well-being (not only the well-being of residents, but that of all humanity) [2]. Since the definition of human well-being by the MA (2005) [3], scholars have started paying attention to the intrinsic connection between ecosystems and human well-being. The topic of how to alter the imbalanced state

**Competing interests:** The authors have declared that no competing interests exist

of ecosystems and harmonize the relationship between ecosystem services and human well-being has become a hot research area in academia [4, 5].

Ecosystem services are the source of human well-being, and even subtle changes in ecosystem services can lead to drastic changes in human well-being, which are particularly evident in ecologically vulnerable areas [6]. Hu et al. [7] proposed that ecosystem services contribute to human well-being at different levels. However, when ecosystems are damaged, they have a negative impact on human well-being. Qiu et al. [8] pointed out that the relationship between human well-being and ecosystem services is not a simple linear relationship but rather a complex and bidirectional relationship that exhibits dynamic changes over time and space. In order to better explore the degree of association and spatial evolution mechanisms between ecosystem services and human well-being, some scholars have utilized coupled coordination models to study their relationship. These researches are mostly conducted at the provincial or municipal level [9, 10], and preliminary spatial-temporal analyses of the coupled coordination relationship have been conducted. Some scholars have also investigated the impacts of different influencing factors on the coupled coordination relationship from the perspectives of policy formulation [11], socio-economic development [5], and other aspects. Their research methods mainly include correlation analysis [12], GWR models [13], GTWR models [14], and so on. The coupling coordination between ecosystems and residents' well-being is closely related to the rationality of industrial structure and exhibits geographic spatial interaction effects, which are manifested in the form of "poverty traps" [15]. Industrial structure serves as a crucial link between human activities in modern society and ecosystems [16]. It plays an important role in promoting regional coordinated development, reducing regional economic disparities, and improving residents' quality of life [17]. Industrial structure is an important dimension for studying the development status of industries, and it is typically measured by indicators such as the rationalization and intensification of industrial structure to assess its quality and evolutionary trends [18]. Industrial structure upgrading is a dynamic process that primarily refers to the transition of industrial structure from a lower level to a higher level. Industrial structure rationalization refers to the degree of coordination between industries and the efficient utilization of resources [19]. It is an important indicator for measuring the coupling degree between factor inputs and output structures [20]. Based on the aforementioned research, we can observe that there exists a challenging contradiction between ecological conservation and economic development. To mitigate this contradiction, a viable approach is industrial upgrading. However, achieving a well-founded industrial upgrading necessitates a comprehensive understanding of its impact mechanisms on both ecosystem services and economic development. Currently, there is limited research in the academic community addressing this aspect. Moreover, the spatial econometric model adopted in this study not only enables the exploration of relationships among variables but also investigates the interactive effects between regions. Presently, there is scarce research employing this model to investigate the spatial interaction effects of industrial upgrading. Through quantitative data analysis, it can offer scientific guidance and data support for high-quality and sustainable regional development.

Correctly managing the relationship between the transformation and upgrading of industrial structure in rural counties and the coordinated development of ecosystems and well-being, especially in poverty-stricken rural counties, is of significant importance for ensuring sustainable economic development in counties and building beautiful rural areas [21, 22]. In the 21st century, the production activities in the Sichuan-Chongqing region of China have rapidly developed, and the industrial structure has been continuously improved [23]. People's living standards and sense of happiness have significantly increased while enjoying the benefits of the ecological environment. However, this development has also brought a series of

problems, such as dwindling resources, environmental degradation, declining biodiversity, and large-scale industrial and population agglomeration [24]. The transformation and upgrading of industrial structure can promote the coordinated development of ecological quality and residents' well-being in rural counties, fundamentally breaking through the established model of independent development in a single region and driving the development of impoverished areas. Whether the existing industrial structure in rural counties can match the development of ecosystems and well-being becomes an important aspect of sustainable economic development in the Sichuan-Chongqing region in the new era [25].

This paper is structured as follows: Sect. 2 provides mechanism analysis, variable selection, model setting, and research methods; Sect. 3 presents the empirical testing and results analysis; Sect. 4 concludes and discuss the findings, and provide policy recommendations.

## 2 Variable selection, model setting, and research methods

### 2.1 Mechanism analysis

The research on the coupling and coordination relationship between the well-being of rural residents and ecological quality in counties is essentially the search for a social operating model that promotes harmonious coexistence between humans and nature. These two systems are closely interconnected with complex interactive coupling relationships. Exploring the internal logic between them is the basis for identifying sustainable operating models and addressing some of the challenges in rural development in certain counties (Fig 1).

Improving the quality of life has been the fundamental goal of all human production activities since the establishment of human society. This article measures the development level of well-being among residents in the region from four dimensions: economic vitality, basic material needs, healthcare, and cultural education. Ecosystem services are the benefits that humans directly or indirectly derive from the natural environment, and they can be categorized into four types: provisioning services, regulating services, supporting services, and cultural services. Provisioning services (such as food and raw material production) have the most direct impact on residents' well-being and economic development. They serve as the fundamental guarantee for residents' engagement in economic activities in rural counties. Regulating services (such as climate regulation and hydrological regulation) create suitable environmental conditions for the operation of the rural economy and provide important support for residents' health and

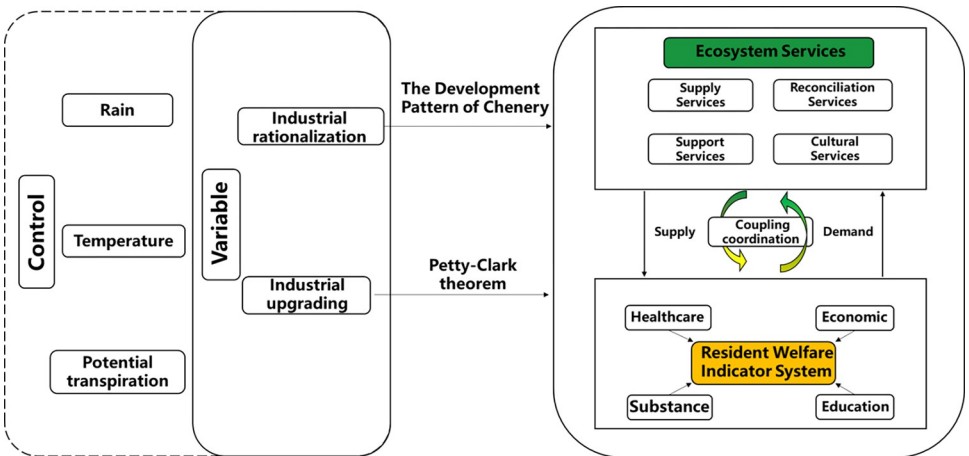

**Fig 1. Analysis framework of the impact mechanism of ecological-well-being coupling coordination.**

safety [26]. Supporting services (such as soil preservation) indirectly influence social-economic development by maintaining nutrient cycles, limiting the exploitation of ecological resources due to potential governance costs. Cultural services (such as providing aesthetic landscapes) provide intangible cultural values for the development of the rural economy [27], promoting the development of cultural education and positive social relationships. The two systems have complex supply-demand relationships, and ecosystem services serve as the original capital for residents' participation in social production. Excessive exploitation of natural resources can lead to unsustainable economic development, where the costs of governance exceed the value of the obtained resources. Exploring the driving factors behind the coupling and coordination degree of ecology and well-being can avoid sacrificing the development of one system at the expense of providing impetus for the development of the other system.

Industrial structural transformation and upgrading is one of the solutions to address the human-land contradiction. Exploring the inherent logic of the impact of industrial structural transformation and upgrading on ecological quality and residents' well-being is a prerequisite for achieving industrial upgrading and sustainable economic development in rural counties. To ensure the rationality and scientific of the construction of the indicator system for industrial structural transformation and upgrading, as well as the selection of influencing factors, this study selects research variables based on existing theories and previous research findings, with the scientific guarantee of whether the model passes the significance level testing at the statistical level. The specific selection of indicators is as follows:

Industrial Rationalization [28]: According to the "H. Chenery development model", a healthy economic system exhibits regular changes in industrial structure as the economy continues to develop, gradually approaching the ideal economic structure during different stages of economic development. Currently, the rural counties in the Sichuan-Chongqing region are undergoing industrial adjustments aimed at reducing the proportion of the primary and secondary sectors while increasing the proportion of the tertiary sector. This can effectively bring the economic structure closer to the ideal standard for economic development. Therefore, in this study, industrial rationalization is used to measure the impact of the level of industrial rationalization in the Sichuan-Chongqing rural counties on the coupling and coordination degree of ecology and well-being.

Industrial Upgrading [28]: According to the Lewis-Clark theorem (which states that as economic development and per capita national income increase, labor first shifts from the primary sector to the secondary sector, and then to the tertiary sector). Economic development-induced industrial structural transformation and upgrading, under the same level of output, can reduce the economy's dependence on natural resources, leading to improved ecological quality and subsequently affecting the coupling and coordination degree of regional ecology and economy. Therefore, in this study, industrial upgrading is used to measure the extent of industrial upgrading and its impact on the coupling and coordination degree of ecology and well-being.

Control Variables [29]: Due to the heterogeneity of the terrain in the Sichuan-Chongqing rural counties, there are fluctuations in rainfall, temperature, and vegetation types during the study period. Therefore, in this study, Temperature, Rain, and Potential transpiration are selected as control variables, and both spatial fixed effects and time fixed effects are introduced into the model.

## 2.2 Regional overview

The Sichuan-Chongqing region (Fig 2), located between 97 degrees 21 minutes to 108 degrees 33 minutes east longitude and 29 degrees 03 minutes to 34 degrees 19 minutes north latitude

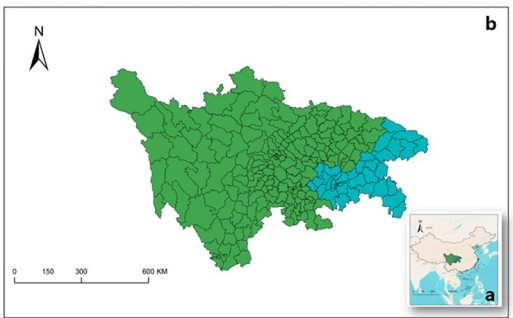
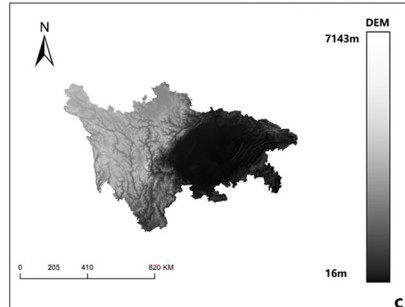

**Fig 2. Administrative division, land use, and elevation map of the rural counties in Sichuan-Chongqing area.** *The image is sourced from (https://www.resdc.cn/), and was created using ArcGIS.

in southwestern China, marks the transitional zone between the Qinghai-Tibet Plateau and the middle-lower reaches of the Yangtze River. Ranging from an elevation of 188 to 7556 meters, the region's topography exhibits remarkable variations, with higher elevations to the west and lower to the east.

The complex landforms are defined by distinct features: the western area of Huaying Mountain-Bayue Mountain consists of hilly terrain, while the region between Huaying Mountain and Fangdou Mountain presents the parallel ridge-valley area in eastern Sichuan, known for its agricultural potential. In the north lies the Zhongshan area of Daba Mountain, a significant ecological zone, while the eastern, southeastern, and southern regions encompass mountainous areas of Wushan and Daluoshan, constituting a pattern referred to as "two hills, seven mountains, and three flat basins" [30].

These geographical features contribute to the diverse vegetation types and rich species resources of the region, affirming its role in biodiversity conservation and national ecological security. The rivers and mountains shape not only the physical landscape but also influence the socio-economic fabric of the rural counties in the upper reaches of the Yangtze River.

The unique position of this region implies that social-economic development is subject to severe ecological constraints. Acting as an ecological barrier in the upper reaches of the Yangtze River, it bears significant environmental pressures, while simultaneously serving as a catalyst for the economic development of the western region [31]. The challenge lies in balancing the ecological well-being with economic growth, as any imbalance can severely hinder sustainable development at the county level [32]. Thus, fostering the transformation and upgrading of the industrial structure in the Sichuan-Chongqing rural areas is vital for enhancing the coordinated development of the region, intricately weaving together the natural elements with social and economic dynamics.

## 2.3 Data sources

The research data mainly came from *the China County Statistical Yearbook*, statistical yearbooks of various cities, *the Agricultural Statistical Yearbook*, 30-meter precision land cover dataset (https://zenodo.org/), Geospatial Data Cloud (http://www.gscloud.cn/), National Meteorological Data Science Center (http://data.cma.cn/), etc. To ensure the validity of the samples, the following data processing was conducted in this study:

1. Exclusion of samples with significant missing key data.

2. For individual samples with missing values, the data was supplemented based on county statistical yearbooks, statistical bulletins, or linear interpolation methods.

3. All the data are open-source, and readers can find the data used on the website.

## 2.4 Research methods

1. Construction of Evaluation Indicator System for Coupling Coordination between Ecological Quality and Rural Resident Well-being

The coupling coordination between ecological quality and rural resident well-being refers to the mutual interaction and influence between the well-being of residents and the ecological system in terms of time scale or spatial scope. It involves the orderly evolution of elements within each system, achieving coordinated development and realizing the common advancement of residents' living standards and environmental protection.

(1) Construction of Evaluation Indicator System for Rural Resident Well-being

The pursuit of a high quality of life by residents is a driving force for social development and a hot topic in today's academic community. Drawing on existing research findings and approaches [10, 33, 34], and considering the criteria of scientific rigor, relevance, and consistency in indicator selection, this study employs the entropy weighting method to assess the overall well-being of rural residents in the Sichuan-Chongqing region from 2010 to 2020. The evaluation is based on four dimensions: economic stability, basic material conditions, healthcare and well-being, and cultural education. The specific indicators are presented in Table 1.

Economic Vitality Indicators: The quality of residents' lives is built upon a foundation of economic stability and vitality. In this study, the economic stability and development dynamics of the region are primarily assessed through indicators such as general public revenue, general public expenditure, per capita GDP, and urbanization rate. These indicators serve as measures of economic stability and growth potential in the area.

Basic Material Indicators: Basic material resources are essential for the living standards of rural residents. In this study, the level of basic material resources available or potentially accessible to the rural counties is measured using indicators such as effective irrigation area, road mileage, per capita grain production, fertilizer application rate, rural electricity consumption, and vegetation coverage. These indicators serve as measures of the level of basic material resources that are available or potentially accessible to the rural counties.

Healthcare Indicators: The level of healthcare plays a crucial role in the well-being of rural residents. In this study, the level of healthcare is measured using indicators such as the number of hospital and clinic beds, the number of beds in social welfare institutions, the number of social welfare institutions, the total number of hospitals, clinics, and doctors in the region. These indicators serve as measures of the healthcare level in the rural counties.

Cultural and Educational Indicators: The level of cultural and educational development reflects the access to educational resources and is an important component of the local talent cultivation and development potential. In this study, the cultural and educational well-being of rural residents is measured using indicators such as the number of primary and secondary schools and the average student-to-teacher ratio in primary and secondary schools. These indicators provide insights into the cultural and educational well-being of residents in the rural counties.

Finally, the indicators are integrated into the Rural Resident Well-being Index through the entropy weighting method.

(2) Ecological quality value assessment

Ecological quality refers to the natural environmental conditions and benefits that ecosystems form and maintain, which are essential for human survival. Assessing the ecological

**Table 1. Evaluation indicator system table for residents' well-being system.**

| Criterion Level | Indicator Level | Indicator Calculation |
|---|---|---|
| Economic vitality -0.2441 | General fiscal revenue(0.1140) | Local fiscal revenue (thousand yuan) (+) |
| | General fiscal expenditure(0.0570) | Local fiscal expenditure (thousand yuan) (+) |
| | Per capita GDP(0.0381) | Gross Domestic Product (GDP) per capita (in yuan/year) (+) |
| | Urbanization rate(0.0320) | Urban population/Total population (+) |
| Basic material needs -0.1846 | Effective irrigation area(0.1059) | Annual actual irrigated area (in hectares) (+) |
| | Road mileage(0.0464) | Total road mileage at the end of the year (in kilometers) (+) |
| | Per capita grain production(0.0309) | Annual grain production/Total population (in tons/person) (+) |
| | Fertilizer application rate(0.0740) | Annual total fertilizer usage (in tons) (+) |
| | Rural electricity consumption(0.0756) | Annual cumulative rural electricity consumption (in kilowatt-hours) (+) |
| | Vegetation coverage(0.0041) | Vegetation coverage area/ Total area (+) |
| Healthcare -0.2956 | Number of hospital and clinic beds(0.0580) | Total number of hospital beds at the end of the year (+) |
| | Number of social welfare institutions(0.0797) | Number of social welfare institutions at the end of the year (+) |
| | Number of hospitals and clinics(0.0255) | Number of hospitals and health clinics at the end of the year (+) |
| | Number of social welfare institution beds (0.0652) | Number of beds in social welfare institutions at the end of the year (+) |
| | Number of doctors(0.0672) | Number of doctors at the end of the year (+) |
| Cultural education -0.1294 | Number of regular secondary schools(0.0537) | Number of secondary schools at the end of the year (+) |
| | Number of regular primary schools(0.0728) | Number of primary schools at the end of the year (+) |
| | Average student-to-teacher ratio in primary schools(0.0006) | Number of primary school students per primary school teacher (-) |
| | Average student-to-teacher ratio in junior high schools(0.0023) | Number of secondary school students per secondary school teacher (-) |

quality through the valuation of ecosystem services not only helps establish a foundation for ecological functional zoning and ecological construction planning but also provides scientific basis for the delineation of ecological functional zones and the planning of ecological construction. In order to explore the ecological development situation in the Sichuan-Chongqing rural areas, this paper adopts the value equivalent method proposed by Norah [35]. to assess the ecosystem service value of Sichuan-Chongqing rural areas from 2010 to 2020, and makes adjustments to the economic value of equivalent factors [36], using ecosystem service value as a measure of ecological quality development. The specific assessment method is as follows:

$$E = \frac{1}{7} \times (P \times Y) \times q \qquad (1)$$

$$ESV = \sum_{j=1}^{m} \sum_{i=1}^{n} A_i E_{ij} \tag{2}$$

$$AESV = \frac{ESV = \sum_{j=1}^{m} \sum_{i=1}^{n} A_i E_{ij}}{\sum_{i=1}^{n} A_i} \tag{3}$$

In the equation, $E$ represents the ecosystem service value per unit area; $P$ represents the average price of grain; $Y$ represents the average yield of grain crops per unit area; $q$ represents the value equivalent coefficient; $ESV$ represents the total value of ecosystem services; $A_i$ represents land use type; and $AESV$ represents the per capita ecosystem service quantity. Among them, the per unit area ecosystem service value in the Yellow River Basin based on the correction coefficient is shown in Table 2 as follows:

In addition, due to the differences in the nature and standards of the indicators, direct comparisons are not possible. Therefore, this study used the range method to normalize the indicators $ESV$ on a per unit area basis for the years 2010–2020.

$$PESV = \frac{PESV_i - PESV_{\min}}{PESV_{\max} - PESV_{\min}} \tag{4}$$

In the equation, $PESV$ represents the per $ESV$ unit area; $PESV_i$ represents the per $ESV$ unit area of the region $i$; $PESV_{\max}$ and $PESV_{\min}$ and represent the maximum and minimum values of the per $ESV$ unit area within the study area respectively.

**Table 2. Ecosystem service value table for rural areas in Sichuan and Chongqing (CNY/hectare).**

| Primary Land Use Type | Secondary Land Use Type | Forest | Grassland | Farmland |
|---|---|---|---|---|
| Supply services | Food production | 888.756 | 1158.076 | 2693.2 |
| | Raw material production | 8025.736 | 969.552 | 1050.348 |
| Regulating services | Gas regulation | 11634.62 | 4039.8 | 1939.104 |
| | Climate regulation | 10961.32 | 4201.392 | 2612.404 |
| | Hydrological regulation | 11015.19 | 4093.664 | 2073.764 |
| | Waste management | 4632.304 | 3555.024 | 3743.548 |
| Supporting services | Soil conservation | 10826.66 | 6032.768 | 3959.004 |
| | Maintaining biodiversity | 12146.33 | 5036.284 | 2747.064 |
| Cultural services | Providing aesthetic landscapes | 5601.856 | 2343.084 | 457.844 |
| | Total | 75732.78 | 31429.64 | 21276.28 |
| Primary Land Use Type | Secondary Land Use Type | Wetland | Waterbody | Desert |
| Supply services | Food production | 969.552 | 1427.396 | 53.864 |
| | Raw material production | 646.368 | 942.62 | 107.728 |
| Regulating services | Gas regulation | 6490.612 | 1373.532 | 161.592 |
| | Climate regulation | 36492.86 | 5547.992 | 350.116 |
| | Hydrological regulation | 36196.61 | 50551.36 | 188.524 |
| | Waste management | 38782.08 | 39994.02 | 700.232 |
| Supporting services | Soil conservation | 5359.468 | 1104.212 | 457.844 |
| | Maintaining biodiversity | 9937.908 | 9237.676 | 1077.28 |
| Cultural services | Providing aesthetic landscapes | 12631.11 | 11957.81 | 646.368 |
| | Total | 147506.6 | 122136.6 | 3743.548 |

(3) The Ecological-Wellbeing Coupling Coordination Evaluation Model

The Ecological-Wellbeing Coupling Coordination Evaluation Model analyzes the relationship between residents' quality of life and the coordination of ecological development. It uses the Coupling Coordination Degree Index to measure the degree of coordination between the two factors. Coupling degree refers to the interaction between different systems, while coordination degree reflects whether different systems are developing at the same level. The focus of this study is on the coordinated development of ecology and well-being. When both grow simultaneously, the result of the coupling coordination degree will also increase, meaning that they become more coordinated. This paper constructs the Ecological-Wellbeing Coupling Coordination Evaluation Model based on the indices of ecosystem services and residents' well-being system. The specific details are as follows:

$$C = \frac{2\sqrt{U_1 U_2}}{U_1 + U_2} \tag{5}$$

$$T = \alpha U_1 + \beta U_2 \tag{6}$$

$$D = \sqrt{C \times T} \tag{7}$$

In the equation, $C$ represents the coupling degree of the systems. A higher value of $C$ indicates a higher level of coupling between the two systems, while a lower value indicates a lower level of coupling. $U_1$ and $U_2$ represent the indices of ecosystem services and residents' well-being. $T$ represents the comprehensive coordination index between residents and the ecosystem system. $\alpha$ and $\beta$ represent the weights assigned to each system in the coordination degree model. Since ecological quality and residents' well-being are equally important, so $\alpha = \beta = 0.5$. $D$ represents the coupling coordination degree. Additionally, this paper refers to the classification criteria for the levels of ecological-wellbeing coupling coordination obtained from Peng [37].

2. Analysis of Factors Influencing Ecological-Wellbeing Coupling Coordination Degree

(1) Industrial Rationalization

This article uses the industrial rationalization to vertically examine the "quality" requirements in the process of industrial structure transformation and upgrading. It also employs the advancement of industrial upgrading examine the "quantity" requirements in the process of industrial structure transformation and upgrading. This approach ensures that the research conclusions are more comprehensive and reliable. Firstly, the industrial rationalization refers to the transformation of industrial structure from imbalance to balance. It involves adjusting the industrial structure while taking into account the sound operation of the social-economic system, ultimately achieving an upgrade in terms of "quality". The Theil index provides a good solution framework, and its formula is as follows:

$$TL = \sum_{i=1}^{n} (\frac{Y_i}{Y}) \ln(\frac{Y_i/L_i}{Y/L}) \tag{8}$$

In China, there are three major industries, hence $n$ denoted as 3. $i$ representing specific industries, $L$ representing employment numbers, and $Y$ representing industrial output values. Since $TL$ is a reverse indicator, the smaller the result, the more rational the industrial structure; a larger result indicates less rationality.

(2) Industrial Upgrading

Industrial upgrading refers to the process of changing the dominant industries and establishing a higher-level industrial structure. It is specifically reflected in the transition of dominant industries and changes in employment structure, indicating the improvement of industrial transformation in terms of "quantity". In calculating this index, the formula for calculating the industrial structure advancement index is as follows [38]:

$$\theta_j = \arccos\left(\frac{\sum_{i=1}^{3}(x_{i,j} \cdot x_{i,0})}{\left(\sum_{i=1}^{3}(x^2_{i,j})\right)^{1/2} \cdot \left(\sum_{i=1}^{3}(x^2_{i,0})\right)^{1/2}}\right) \tag{9}$$

$$TN = \sum_{k=1}^{3} \sum_{j=1}^{k} \theta_j \tag{10}$$

In the given equation, $i$ represents specific industries. $X_{i,0}$ is a set of three-dimensional vectors formed by the ratio of the $i$ industry (tertiary industry) to GDP. $\theta_j$ represents the angle formed by the vector $X_{i,0}$ and vectors of each industry, where $j$ = 1,2,3 is the angle between them. According to the above formula, a higher value of $TN$, calculated using the given formula, indicates a higher level of industrial structure advancement.

(3) Spatial Durbin Model

The Spatial Durbin Model is a significant extension of the traditional panel model, incorporating both the spatial lag of the dependent variable and spatial autocorrelation coefficients, enabling a comprehensive analysis of spatial interaction effects and impact effects. This utilization of the model reflects the complex spatial interdependence and connectivity in modern economic and social systems. Given the research objective of this paper, which aims to reveal the underlying mechanisms of how industrial transformation and upgrading can alleviate the inherent conflicts between local residents and the environment, this advanced analytical tool was specifically chosen. By constructing a fixed-effects Spatial Durbin Model, this study not only delves into the factors influencing the ecological-wellbeing coupling coordination degree in the rural counties of the Sichuan-Chongqing region but also meticulously examines the spatial interaction effects among these factors, uncovering deeper dynamic relationships. Such an approach not only enhances the precision and sensitivity of the analysis but may also provide a more solid foundation for future policy-making. The specific details are as follows:

$$Y_{it} = \rho W Y_{it} + \beta X_{it} + \lambda W X_{it} + u_1 + u_2 + \varepsilon_{it} \tag{11}$$

In the equation, $Y_{it}$ represents the dependent variable in the model. $\mu_1$ and $\mu_2$ represent the time fixed effects and spatial fixed effects. $i$ and $t$ represent the regional and yearly variables. $X_{it}$ is the set of all independent variables. $W$ is the spatial weight matrix. $\varepsilon_{it}$ represents the error term. $\rho$, $\beta$ and $\lambda$ are coefficients.

## 3 Empirical testing and results analysis

### 3.1 Analysis of spatiotemporal evolution of ecological-wellbeing coupling coordination degree in the rural counties of Sichuan-Chongqing region

To explore the coordinated development of ecological quality and resident well-being in the rural counties of the Sichuan-Chongqing region during the period from 2010 to 2020, this section takes a dual perspective of time and space to reveal the evolving trends of ecological-wellbeing coupling coordination degree in the region. The specific analysis is as follows:

(1) Time Evolution Analysis

**Table 3. The classification of coordination degree in the economic-ecological coupling coordination model.**

| coordination degree | 0–0.09 | 0.10–0.19 | 0.20–0.29 | 0.30–0.39 | 0.40–0.49 |
|---|---|---|---|---|---|
| coordination level | extreme imbalance | severe imbalance | moderate imbalance | mild imbalance | on the brink of imbalance |
| coordination degree | 0.50–0.59 | 0.60–0.69 | 0.70–0.79 | 0.80–0.89 | 0.90–1.00 |
| coordination level | tenuously coordinated | primary coordination | intermediate coordination | good coordination | high-quality coordination |

Based on Table 3, this paper presents the division of coordination levels and depicts the changes in the ecological-wellbeing coupling coordination rate and average coupling coordination degree in the rural counties of the Sichuan-Chongqing region (Fig 3). Additionally, it includes a Sankey diagram illustrating the transfer matrix of ecological-wellbeing coupling coordination degree in the region (Fig 4). In Fig 3, the blue line represents the average coupling coordination degree of all rural counties in the Sichuan-Chongqing region, while the green bars represent the coordination rate (the percentage of counties with a coupling coordination degree higher than 0.5 among all counties). During the study period, the average coupling coordination degree of the counties in the region increased from 0.4648 to 0.4997, with a total growth of 0.0325, indicating a growth rate of 7.5% and an average annual growth rate of 0.68%. The overall trend showed fluctuating growth. The coordination rate, which was 31.36% in 2010, rose to 48.64% in 2020. Over the span of 11 years, it increased by 17.28% with an average annual growth rate of 1.57%. The period from 2010 to 2017 experienced an accelerated growth phase, showing an exponential growth rate. From 2017 to 2020, the growth rate slowed down, gradually approaching convergence, and the overall coordination rate approached 50%, indicating that nearly half of the regions reached a coordinated level.

In Fig 4, the data on the left side (eg: 5, 22, 38, and 1) represent the total number of transitions from low coordination levels to high coordination levels in the rural counties of the Sichuan-Chongqing region from 2010 to 2020. On the right side, it represents the total number of transitions from high coordination levels to lower coordination levels. Most of the transitions

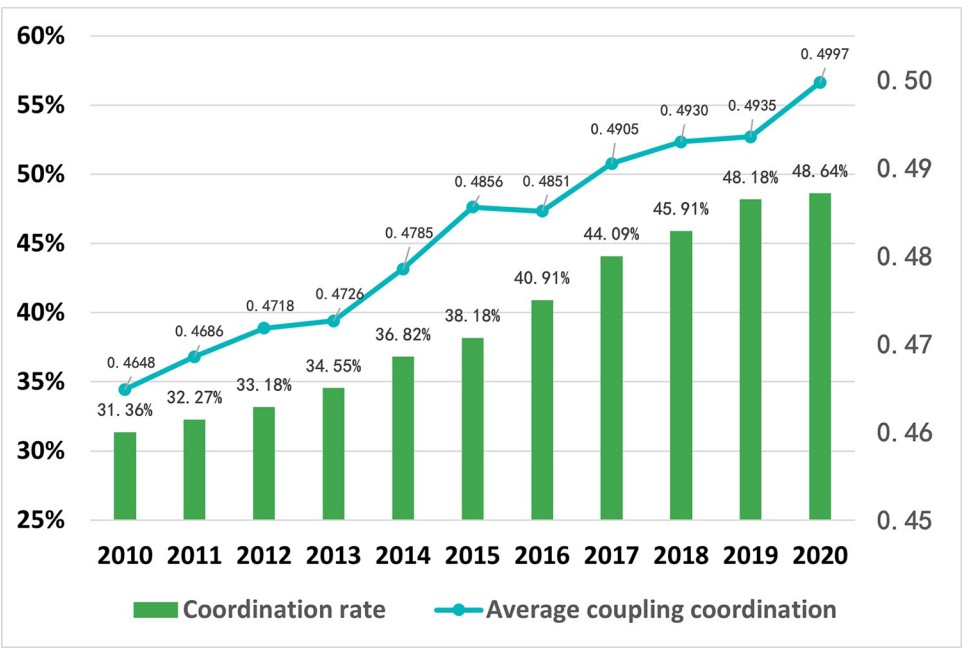

**Fig 3. Chart of coordination rate and average coupling coordination changes in the rural counties of the Sichuan-Chongqing region from 2010 to 2020.**

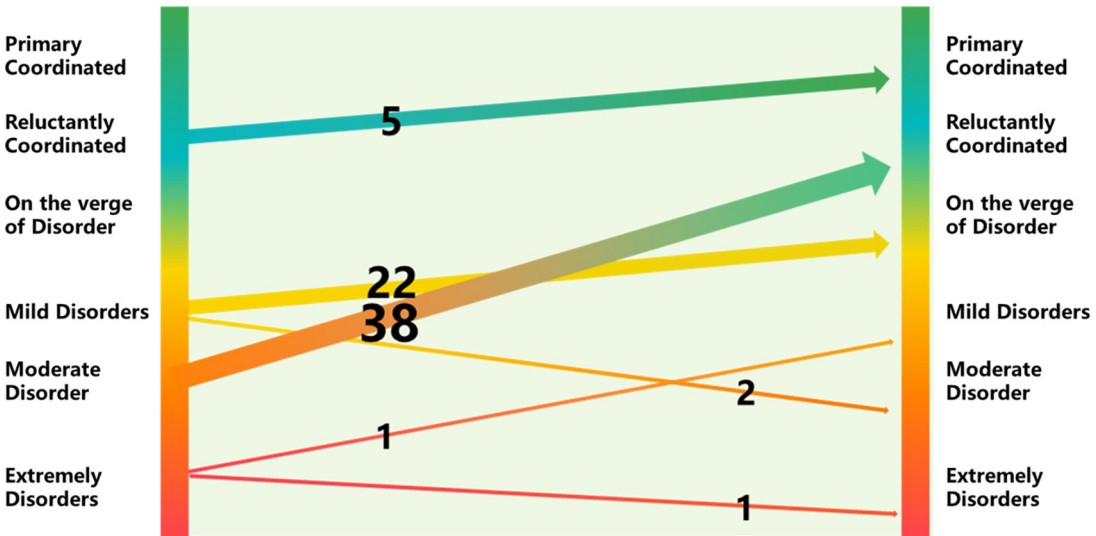

**Fig 4. Sankey diagram of transition matrix in the rural counties of the Sichuan-Chongqing region from 2010 to 2020.**

indicate an improvement in the coordination level of the counties, with only three counties experiencing a decrease in coordination level over the 11-year period. Out of the total transitions, 38 counties (townships) improved from being on the verge of disarray to achieving weak coordination, accounting for 55% of the total transitions. Twenty-four counties (townships) transitioned from a mild state of imbalance to other coordination levels. Twenty-two counties (townships) transitioned to a state of imminent disarray. Two counties (townships) were in a state of moderate imbalance. Five counties (townships) underwent significant optimization, transitioning from weak coordination to primary coordination. Only one county (township) experienced a continuous deterioration in coordination level, transitioning from moderate imbalance to extreme imbalance. Overall, the coordination level of the rural counties in the Sichuan-Chongqing region has shown improvement, with only three counties (townships) experiencing a decline in coordination level, while 66 counties (townships) have seen varying degrees of improvement in their coordination level.

(2) Spatial evolution analysis

This article selects the ecological-well-being coupling and coordination degree in the rural areas of Sichuan and Chongqing provinces in 2010, 2015, and 2020. The spatial evolution trend map is drawn using ArcGIS, as shown below (Fig 5). From the graph, it can be observed

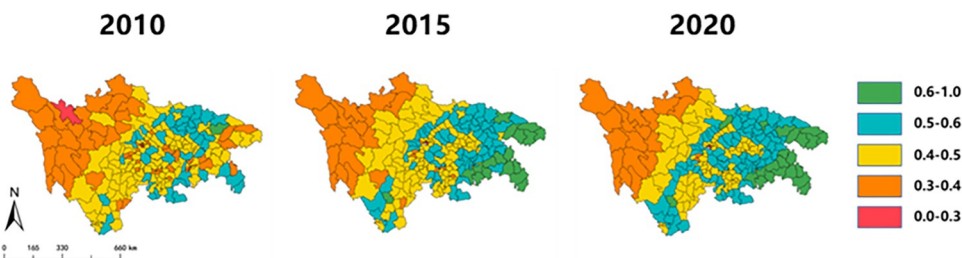

**Fig 5. The spatial evolution trend map of the ecological-well-being coupling and coordination degree in the rural areas of Sichuan and Chongqing provinces in 2010, 2015, and 2020.** *The image is sourced from (https://www.resdc.cn/), and was created using ArcGIS.

that the spatial distribution of the coupling and coordination degree in the rural areas of Sichuan and Chongqing provinces shows a pattern of higher values in the east and lower values in the west, indicating significant heterogeneity. Over the course of 11 years, there were substantial differences in the growth rates among different regions. The eastern region of Chongqing experienced the highest growth rate, with some townships and counties shifting from the range of (0.3–0.5) to (0.6–1.0). According to the transition matrix in Fig 4, most of the areas that transitioned from being on the brink of imbalance to barely coordinated counties or townships were concentrated in the eastern region of Chongqing. In the central region, there were numerous areas that transitioned from the range of (0.4–0.5) to (0.5–0.6), representing a shift from mild imbalance to basic coordination, mainly concentrated in the central region of Sichuan and Chongqing. The western region remained at a relatively low level of coordinated development over the 11-year period.

Considering the land use distribution in Fig 1, it can be observed that the western region has higher altitudes and is mainly covered by forests and grasslands. It also has lower levels of transportation infrastructure, and the quality, quantity, and industrial development level of the population need continuous improvement. In contrast, the eastern and central regions are mainly flat and dominated by arable land and built-up areas. They have the basic conditions for the formation of industrial clusters. Therefore, the pace of industrial structural transformation in the eastern and central regions is faster than in the western region, resulting in the spatial pattern of higher values in the east and lower values in the west, as well as the trend of faster growth in the east and slower growth in the west.

## 3.2 The impact analysis of industrial transformation and upgrading in the rural areas of Sichuan and Chongqing provinces on the coupling and coordination degree of ecology and well-being

To further explore the impact mechanism of industrial structural transformation and upgrading on the coupling and coordination degree of ecology and well-being in the rural areas of Sichuan and Chongqing provinces, this study utilizes Formulas (8–10) to calculate the indices of industrial rationalization and industrial advancement as measures of the status of industrial structural transformation and upgrading. The rationalization and advancement indices are used as explanatory variables, while the coupling and coordination degree of ecology and well-being is taken as the dependent variable. Additionally, potential evapotranspiration, precipitation, and temperature are selected as control variables (Table 4) to construct an OLS (Ordinary Least Squares) model.

(1) Spatial Correlation Analysis and Model Selection

Based on the ecological-well-being coupling and coordination degree in the rural areas of Sichuan and Chongqing provinces, an OLS regression model has been constructed. However, there is a possibility of spatial auto-correlation in the residuals of the model. In this study, the

**Table 4. Factors influencing the coupling and coordination degree of ecology and well-being in the rural areas of Sichuan and Chongqing provinces.**

| Variable Types | Influencing Factors | Unit | Mean | Standard Deviation |
|---|---|---|---|---|
| Explanatory Variables | Industrial Rationalization | | 0.442 | 0.33 |
| | Industrial Upgrading | | 6.274 | 0.412 |
| Control Variables | Potential Evapotranspiration | mm | 1011 | 112.1 |
| | Precipitation | mm | 1096 | 250.1 |
| | Temperature | ˚C | 13.5 | 5.6 |

**Table 5. The Moran's I test.**

| Year | 2010 | 2011 | 2012 | 2013 | 2014 | 2015 | 2016 | 2017 | 2018 | 2019 | 2020 |
|---|---|---|---|---|---|---|---|---|---|---|---|
| Moran I | 0.0987 | 0.0898 | 0.0856 | 0.1003 | 0.1111 | 0.1020 | 0.1037 | 0.1188 | 0.1073 | 0.1098 | 0.1188 |
| t | 12.45 | 11.44 | 10.96 | 12.63 | 13.88 | 12.79 | 13.05 | 14.77 | 13.47 | 13.78 | 14.38 |
| p | 0 | 0 | 0 | 0 | 0 | 0 | 0 | 0 | 0 | 0 | 0 |

Moran I test was conducted on the residuals of the model using MATLAB 2016 and the jplv7 spatial econometrics package developed by Elhorst. The results of the Moran I test are presented in Table 5:

Based on the information provided, the Hausman test was used to determine whether fixed effects or random effects should be used in the model. Additionally, the LM (Lagrange Multiplier) test was conducted to identify the type of spatial effects present in the model. The results, as shown in Table 6, indicate that the OLS regression model is best represented by the Spatial Durbin Model (SDM) with double fixed effects. This model is used to analyze the influencing factors and spatial interaction effects on the coupling and coordination degree of ecology and well-being in the rural areas of Sichuan and Chongqing provinces at the township and county scale.

(2) Model Results Analysis

Based on Table 6, this study constructs a double fixed spatial Durbin model, and the regression results are as follows (Table 7):

1. Ecological-Well-being Coupling Spillover Effect: The coefficient of the coupling degree $\rho$ is significant and positive. An increase in the coupling degree in one region substantially enhances the coupling degree in the surrounding townships and counties, reflecting strong policy transmission and agglomeration effects in the rural areas of Sichuan and Chongqing. When these areas enhance their coordination level, neighboring regions with similar social structures and conditions benefit, ultimately improving the overall ecological-well-being coupling and coordination degree.

2. Industrial Rationalization: Both the direct and indirect effects of industrial rationalization are significantly negative at a 1% level. This suggests that a decrease in the industrial rationalization index (indicating less rationalized industries) is associated with an increase in the ecological-well-being coupling and coordination degree of the local and surrounding townships and counties. The elasticities for the indirect effects are much larger than the direct effects (0.0593 and 2.5248). This implies that a more rationalized industry in the local area and its surrounding townships and counties leads to higher coordination levels. The impact of industrial rationalization on the surrounding areas is much greater than its impact on the local area. This can be attributed to the similarity in culture [39], strong population and capital mobility, and the high transmission of policies, social structures, and development

**Table 6. Results of the various tests.**

| | Spatial Fixed Effects | Time Fixed Effects | LM-error | LM-lag | robust LM-error |
|---|---|---|---|---|---|
| Coefficients | 275.71 | 927.49 | 27.41 | 903.97 | 876.58 |
| P | 0 | 0 | 0 | 0 | 0 |
| | robust LM-lag | LR_sar | LR_sem | Wald_sar | Wald_sem |
| Coefficients | 0.01 | 22.89 | 41.79 | 33.93 | 48.22 |
| P | 0.89 | 0 | 0 | 0 | 0 |

**Table 7. The regression results of the model.**

| Variable Name | OLS | SDM | Direct Effect | Indirect Effect | Total Effect |
|---|---|---|---|---|---|
| Industrial Rationalization | -0.0826*** | -0.0503*** | -0.0593*** | -2.5248*** | -2.5841*** |
| Industrial Upgrading | 0.0269*** | 0.0182*** | 0.0159*** | -0.7275** | -0.7116** |
| Potential Evapotranspiration | -0.0001*** | 0.0001 | 0.0001 | 0.0044 | 0.0045 |
| Precipitation | 0.0001*** | 0.0001 | -0.0001 | -0.0002** | -0.0002** |
| Temperature | 0.0023*** | -0.0078 | -0.0156 | -2.1131*** | -2.1131*** |
| W*Industrial Rationalization | | -0.0704*** | | | |
| W*Industrial Upgrading | | -0.0508*** | | | |
| W*Potential Evapotranspiration | | 0.0001 | | | |
| W*Precipitation | | -0.0002*** | | | |
| W*Temperature | | -0.0915*** | | | |
| Con | 0.0519* | | | | |
| ρ | | 0.9519*** | | | |
| R² | 0.3127 | 0.655 | | | |

Note:

***, **, * denote statistical significance at the 1%, 5%, and 10% levels. W represents the spatial weight matrix based on geographic distance, con represents the constant term, and ρ represents the spatial auto-correlation coefficient.

patterns within the rural areas of Sichuan and Chongqing. As a result, the cumulative effect on the surrounding townships and counties, which have similar social structures and scales, is greater than the effect on the local township or county. Additionally, the transition of industrial structure in developing countries follows a common pattern [40]: priority is given to the development of agriculture and industry, followed by the optimization of surplus capacity and human resources in the tertiary sector. When the rural areas of Sichuan and Chongqing undergo the transition from the primary and secondary sectors to the tertiary sector, the industrial rationalization index decreases, leading to increased labor value-added and improved well-being for residents. The increased share of the tertiary sector also reduces the intensity of resource exploitation, gradually improving ecological quality and thereby enhancing the ecological-well-being coupling and coordination degree.

3. Industrial Upgrading: The direct effect of industrial upgrading is significantly positive (0.0159), while the indirect effect (-0.7275) and total effect (-0.7116) are significantly negative. This indicates that a higher industrial upgrading index in a region is associated with a higher ecological-well-being coupling and coordination degree in the local area but a lower coupling degree in the surrounding townships and counties. The elasticity of the indirect effect is greater than that of the direct effect, similar to the reason behind the industrial rationalization results. The presence of a "brain drain" effect in high-paying technological industries in the local area is a factor causing the opposite effects of the direct and indirect effects [41]. A single township or county cannot develop high-tech industries with high technological levels. It requires the absorption of human capital and technological capital from the surrounding townships and counties to promote industrial upgrading and increase the value-added of products. High-tech industries also have a lower environmental impact, contributing to the increase in the ecological-well-being coupling and coordination degree. The surrounding townships and counties, which lose talent, capital, and technology, lack the necessary conditions for industrial upgrading and can only engage in non-high-tech production, leading to a lower level of coordinated development.

## 4 Discussion, conclusions, and policy recommendations

### 4.1 Discussion

In this study, we employed a coupling coordination model to systematically evaluate the spatiotemporal evolution of ecological quality and residents' well-being coordination in the rural counties of the Sichuan-Chongqing region from 2010 to 2020. The findings revealed that there existed an east-west disparity in the level of coordination development within the region, with the eastern part experiencing faster growth while the western part exhibited slower progress. This observation aligns with the conclusions drawn by Wang [42], which can be attributed to variations in the overall regional governance investment. By constructing a dual fixed spatial Durbin model, we delved into the mechanisms and spatial interaction effects of industrial transformation and upgrading in the rural counties of the Sichuan-Chongqing region. Our analysis unveiled a robust spatial correlation within the region, and the coupling coordination of ecological well-being showcased a significant positive spatial spillover effect. This phenomenon could be attributed to the pronounced similarity in local cultures and the frequent interactions between regions, such that changes initiated in one area were often emulated by neighboring ones. The structural changes in one county indeed exerted considerable influence on the surrounding counties, facilitated by the common cultural, economic, and developmental patterns in the region. Both industrial rationalization and industrial upgrading exhibited robust spillover effects, though their mechanisms differed, leading to contrasting spatial spillover effects. Successful local transformation instances could be diffused to neighboring counties, enhancing the degree of ecological-well-being coordination in those areas. However, owing to limitations in technology, capital, and talent resources, only a subset of counties in the Sichuan-Chongqing region possessed the fundamental prerequisites for industrial upgrading. This process might trigger a suction effect, resulting in the outflow of talent, capital, and technology from the surrounding counties and thereby diminishing the well-being of their residents. However, this study also has some limitations, including:

1. The correction coefficient for the value-equivalent coefficient of agricultural prices in this study was based on the average agricultural prices in the Sichuan-Chongqing rural counties, rather than separately calculating the correction coefficients for the two regions, which may reduce the accuracy of the correction.

2. The selected study area focused on the rural counties of the Sichuan-Chongqing region, and due to limited data availability, the construction of the residents' well-being index system only considered economic, material, cultural, and medical aspects.

3. This study used a double fixed spatial Durbin model to explore the influence mechanism of industrial structural transformation and upgrading on the ecological-well-being coupling coordination in the Sichuan-Chongqing rural counties, but did not consider the spatial heterogeneity of the coordination and the obtained model represents an average model for the entire region.

### 4.2 Conclusions

Against the backdrop of industrial transformation in China, exploring methods to break away from unsustainable and low value-added production models is crucial for regions across the country. Quantitative research on the coordinated development and the impact mechanism of industrial structure on local coordination development in the rural counties of the Sichuan-Chongqing region is particularly important. The main conclusions of this study are as follows:

1. From 2010 to 2020, the ecological-well-being coupling coordination in the rural counties of the Sichuan-Chongqing region showed overall stable growth. The coordination index exhibited an east-west disparity, with a faster growth rate in the east and a slower growth rate in the west. The east-west difference was significant and increased over the years.

2. The ecological-well-being coupling coordination in the rural counties of the Sichuan-Chongqing region exhibited a strong positive spatial spillover effect. Priority should be given to developing certain areas, allowing them to utilize the spillover benefits and maximize policy effectiveness.

3. The more rational the industrial structure, the higher the level of coordinated development in both the local and surrounding counties. A higher index of industrial upgrading indicated a better level of local coordination development but a lower level of coordination development in the surrounding areas.

### 4.3 Policy recommendations

Based on the analysis and conclusions of this study, the following policy recommendations are proposed:

1. The spatial distribution of ecological-well-being coordinated development in the rural counties of the Sichuan-Chongqing region is uneven, and significant spatial spillover effects exist. It is necessary to adjust development priorities and focus on improving the coordination development level of certain counties in the western region of Sichuan and Chongqing (such as Luding County, Daofu County, etc.). These counties can serve as demonstration counties for the surrounding rural counties, driving their coordinated development together.

2. The more rational the industrial structure in the rural counties of the Sichuan-Chongqing region, the higher the level of coordinated development. This indicates that there is a significant presence of agricultural and industrial structures in the study area. Therefore, appropriate support should be given to the development of the tertiary industry in rural counties, ensuring that they have the basic conditions for industrial structure transformation and upgrading. This will improve the well-being of rural residents, reduce environmental resource demands, and enhance the level of coordinated development in rural counties.

3. The phenomenon of industrial upgrading in the rural counties of the Sichuan-Chongqing region having a suction effect is observed. This is primarily due to the overall shortage of talent, funds, and technology reserves in the entire Sichuan-Chongqing rural county region, resulting in the need to sacrifice the potential for industrial upgrading and coordinated development in surrounding rural counties. The government needs to increase financial investment, focus on talent cultivation and the introduction of advanced technology, and enable each county in the Sichuan-Chongqing rural county region to have the basic conditions for industrial upgrading.

### Supporting information

**S1 File.**
(XLSX)

### Author Contributions

**Conceptualization:** Fan Yang.

**Data curation:** Fan Yang.

**Formal analysis:** Fan Yang.

**Investigation:** Fan Yang.

**Methodology:** Fan Yang.

**Project administration:** Fan Yang.

**Resources:** Wanlin Qi.

**Software:** Wanlin Qi.

**Supervision:** Jiaqi Han.

**Validation:** Jiaqi Han.

**Visualization:** Jiaqi Han.

**Writing – original draft:** Fan Yang.

**Writing – review & editing:** Fan Yang.

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
