## [Decision Letter · Decision Letter 0]

25 Jul 2023

PONE-D-23-20262Research on the mechanism of promoting coordinated development of ecological well-being in rural counties through industrial transformationPLOS ONE

Dear Dr. Yang,

Thank you for submitting your manuscript to PLOS ONE. After careful consideration, we feel that it has merit but does not fully meet PLOS ONE’s publication criteria as it currently stands. Therefore, we invite you to submit a revised version of the manuscript that addresses the points raised during the review process.

We look forward to receiving your revised manuscript.

Kind regards,

Fuyou Guo, (Ph.D.

Academic Editor

PLOS ONE

Journal Requirements:

   "The funders had no role in study design, data collection and analysis, decision to publish, or preparation of the manuscript"

5. We note that Figures 2 and 5 in your submission contain map/satellite images which may be copyrighted. All PLOS content is published under the Creative Commons Attribution License (CC BY 4.0), which means that the manuscript, images, and Supporting Information files will be freely available online, and any third party is permitted to access, download, copy, distribute, and use these materials in any way, even commercially, with proper attribution. For these reasons, we cannot publish previously copyrighted maps or satellite images created using proprietary data, such as Google software (Google Maps, Street View, and Earth). For more information, see our copyright guidelines: http://journals.plos.org/plosone/s/licenses-and-copyright.

a. You may seek permission from the original copyright holder of Figures 2 and 5 to publish the content specifically under the CC BY 4.0 license.  

6. Please ensure that you refer to Figure 2 in your text as, if accepted, production will need this reference to link the reader to the figure.

Additional Editor Comments:

Reviewer 1 Comments:

A dual-fixed-space Durbin model is constructed to analyze the influence mechanism and spatial interaction effects of industrial transformation and upgrading on the coordination of ecology and well-being. The paper discussed the mechanism of promoting coordinated development of ecological wellbeing in rural counties through industrial transformation.

The structure of the article is clear and the process is explained well here. However, it could be improved in some detail.

1.The abstract can be further refined. For example, the research results in the abstract can be summarized into three points.

2.The contributions of this paper require further clarification and improvement. It is suggested to further improve the innovation points.

3.Some of the statements are written without reference; try to add reference with every statement in your paper.

4. In the discussion part, the author should further explain the reasons for these results.

5. The overall quality of English is good, but need to be checked carefully again. I suggest the authors should look for an English native speaker to further check the language of the paper.

6. Some fresh paper can be used as ref. eg: Liu, F., Sim, J. Y., Sun, H., Edziah, B. K., Adom, P. K., & Song, S. (2023). Assessing the role of economic globalization on energy efficiency: Evidence from a global perspective. China Economic Review, 77, 101897. https://doi.org/10.1016/j.chieco.2022.101897.

7.why nothing in the 3rd part of 2.3 Data sources?

Reviewer 2 Comments:

Abstract Suggestions:

1. The abstract provides a brief introduction to the research background and significance. However, it can be improved by providing more specific information about the research objectives, methodology, and key findings.

2. Clearly explain the research methods and data analysis techniques used in the study, highlighting their strengths and contributions to the research.

3. Include more specific information about the research findings, such as the results of the analysis, significant correlations, and key discoveries.

4. Emphasize the practical implications and policy recommendations derived from the research findings, highlighting their potential impact on promoting sustainable development in rural counties.

1. Introduction Suggestions:

1. Start the introduction by providing a more concise and clear statement of the research objective, clearly stating that this study aims to investigate the mechanisms by which industrial transformation in rural counties promotes the coordinated development of ecological well-being.

2. Provide a comprehensive overview of the challenges faced by rural counties in achieving sustainable development, such as the conflicts between economic growth and environmental protection, and the need to balance industrial development with ecological well-being.

3. Clearly state the research significance and potential contributions of the study, highlighting how the findings can provide valuable insights and guidance for policymakers and practitioners involved in rural development and ecological protection.

4. Provide a brief literature review to highlight the existing gaps in knowledge and how the current study addresses those gaps.

5. Clearly outline the structure of the paper, briefly summarizing the content of each section to give readers an overview of the organization of the article.

2. Variable Selection, Model Setting, and Research Methods Suggestions:

1. Emphasize the importance and relevance of the selected variables to the research topic and objectives, highlighting how they contribute to measuring the ecological well-being and the impact of industrial transformation.

2. Clearly describe the process of constructing the indicator system, including the selection of specific indicators, their measurement methods, and the rationale behind their inclusion.

3. Provide a more detailed explanation of the research methods, such as the coupling coordination evaluation model, spatial analysis techniques, and statistical procedures employed in the study.

4. Highlight the strengths and limitations of the chosen research methods, discussing their suitability for capturing the complex relationships between ecological well-being and industrial transformation.

2.2 Regional Overview Suggestions:

1. Provide a more detailed description of the geographical features and topography of the Sichuan-Chongqing region, including specific characteristics of mountains, rivers, and other natural elements that influence the ecological and socioeconomic dynamics.

2. Highlight the specific ecological constraints and pressures faced by rural counties in the region, such as the degradation of natural resources, environmental protection demands, and the need for ecosystem restoration.

3. Discuss the economic development status and role of the region in promoting economic growth in western China, emphasizing the challenges and opportunities for achieving sustainable development in rural counties.

2.3 Data Sources Suggestions:

1. Provide a more detailed description of the data sources used in the study, including specific datasets, their origins, and reliability.

2. Clearly explain the data processing procedures, including any data cleaning, aggregation, or interpolation techniques employed to ensure the quality and consistency of the data used in the analysis.

3. Discuss the validity and reliability of the data sources, highlighting any potential limitations or biases associated with the data and steps taken to address them.

2.4 Research Methods Suggestions:

1. Provide a more detailed explanation of the construction of the indicator system, including the specific criteria used for selecting indicators and the rationale behind their weighting.

2. Clearly explain the methodology used to evaluate the economic value of ecosystem services and its relevance to assessing ecological quality.

3. Describe the coupling coordination evaluation model in more detail, including the calculation formula, weight allocation, and interpretation of the coupling coordination degree index.

4. Provide a more comprehensive overview of the spatial Durbin model and its relevance to analyzing the spatial effects and interdependencies in the study.

3 Empirical Testing and Results Analysis Suggestions:

1. Provide a more detailed analysis of the temporal evolution of the coupling coordination degree, including specific trends, variations, and significant turning points.

2. Elaborate on the spatial distribution patterns of the coupling coordination degree, discussing any regional disparities and spatial clusters observed in the analysis.

3. Provide a more detailed analysis of the impact of industrial transformation and upgrading on the coupling coordination degree, including specific statistical results, effect sizes, and policy implications.

4. Discuss the implications of the model analysis results, emphasizing the key factors influencing the coordinated development of ecological well-being and providing insights for policymakers and practitioners.

4.1 Discussion Suggestions:

1. Provide a more detailed discussion of the temporal evolution analysis, interpreting the findings in the context of relevant theories and previous research. Discuss the implications of the observed trends and variations in the coupling coordination degree.

2. Discuss the spatial distribution patterns and regional disparities observed in the study, exploring the underlying factors and implications for policy interventions.

3. Highlight the key findings and their significance in advancing the understanding of the mechanisms driving the coordinated development of ecological well-being through industrial transformation.

4.2 Conclusions Suggestions:

1. Clearly summarize the main findings of the study, emphasizing the key factors and mechanisms influencing the coordinated development of ecological well-being in rural counties.

2. Provide a concise statement of the study's contributions and significance, highlighting the implications for sustainable rural development and ecological protection.

3. Offer suggestions for future research directions or areas that warrant further investigation, based on the limitations and knowledge gaps identified in the current study.

4.3 Policy Recommendations Suggestions:

1. Provide specific and actionable policy recommendations based on the research findings, addressing the challenges and opportunities identified in the study.

2. Justify each policy recommendation by linking it to the specific research results and the potential impacts on promoting the coordinated development of ecological well-being.

3. Discuss the practical implications and potential implementation strategies for the proposed policy recommendations, taking into account the local context and stakeholder involvement

Reviewers' comments:

Reviewer's Responses to Questions

**Comments to the Author**

1. Is the manuscript technically sound, and do the data support the conclusions?

Reviewer #1: Partly

Reviewer #2: Yes

2. Has the statistical analysis been performed appropriately and rigorously? 

Reviewer #1: I Don't Know

Reviewer #2: Yes

3. Have the authors made all data underlying the findings in their manuscript fully available?

Reviewer #1: Yes

Reviewer #2: Yes

4. Is the manuscript presented in an intelligible fashion and written in standard English?

Reviewer #1: No

Reviewer #2: Yes

5. Review Comments to the Author

Reviewer #1: Abstract Suggestions:

1. The abstract provides a brief introduction to the research background and significance. However, it can be improved by providing more specific information about the research objectives, methodology, and key findings.

2. Clearly explain the research methods and data analysis techniques used in the study, highlighting their strengths and contributions to the research.

3. Include more specific information about the research findings, such as the results of the analysis, significant correlations, and key discoveries.

4. Emphasize the practical implications and policy recommendations derived from the research findings, highlighting their potential impact on promoting sustainable development in rural counties.

1. Introduction Suggestions:

1. Start the introduction by providing a more concise and clear statement of the research objective, clearly stating that this study aims to investigate the mechanisms by which industrial transformation in rural counties promotes the coordinated development of ecological well-being.

2. Provide a comprehensive overview of the challenges faced by rural counties in achieving sustainable development, such as the conflicts between economic growth and environmental protection, and the need to balance industrial development with ecological well-being.

3. Clearly state the research significance and potential contributions of the study, highlighting how the findings can provide valuable insights and guidance for policymakers and practitioners involved in rural development and ecological protection.

4. Provide a brief literature review to highlight the existing gaps in knowledge and how the current study addresses those gaps.

5. Clearly outline the structure of the paper, briefly summarizing the content of each section to give readers an overview of the organization of the article.

2. Variable Selection, Model Setting, and Research Methods Suggestions:

1. Emphasize the importance and relevance of the selected variables to the research topic and objectives, highlighting how they contribute to measuring the ecological well-being and the impact of industrial transformation.

2. Clearly describe the process of constructing the indicator system, including the selection of specific indicators, their measurement methods, and the rationale behind their inclusion.

3. Provide a more detailed explanation of the research methods, such as the coupling coordination evaluation model, spatial analysis techniques, and statistical procedures employed in the study.

4. Highlight the strengths and limitations of the chosen research methods, discussing their suitability for capturing the complex relationships between ecological well-being and industrial transformation.

2.2 Regional Overview Suggestions:

1. Provide a more detailed description of the geographical features and topography of the Sichuan-Chongqing region, including specific characteristics of mountains, rivers, and other natural elements that influence the ecological and socioeconomic dynamics.

2. Highlight the specific ecological constraints and pressures faced by rural counties in the region, such as the degradation of natural resources, environmental protection demands, and the need for ecosystem restoration.

3. Discuss the economic development status and role of the region in promoting economic growth in western China, emphasizing the challenges and opportunities for achieving sustainable development in rural counties.

2.3 Data Sources Suggestions:

1. Provide a more detailed description of the data sources used in the study, including specific datasets, their origins, and reliability.

2. Clearly explain the data processing procedures, including any data cleaning, aggregation, or interpolation techniques employed to ensure the quality and consistency of the data used in the analysis.

3. Discuss the validity and reliability of the data sources, highlighting any potential limitations or biases associated with the data and steps taken to address them.

2.4 Research Methods Suggestions:

1. Provide a more detailed explanation of the construction of the indicator system, including the specific criteria used for selecting indicators and the rationale behind their weighting.

2. Clearly explain the methodology used to evaluate the economic value of ecosystem services and its relevance to assessing ecological quality.

3. Describe the coupling coordination evaluation model in more detail, including the calculation formula, weight allocation, and interpretation of the coupling coordination degree index.

4. Provide a more comprehensive overview of the spatial Durbin model and its relevance to analyzing the spatial effects and interdependencies in the study.

3 Empirical Testing and Results Analysis Suggestions:

1. Provide a more detailed analysis of the temporal evolution of the coupling coordination degree, including specific trends, variations, and significant turning points.

2. Elaborate on the spatial distribution patterns of the coupling coordination degree, discussing any regional disparities and spatial clusters observed in the analysis.

3. Provide a more detailed analysis of the impact of industrial transformation and upgrading on the coupling coordination degree, including specific statistical results, effect sizes, and policy implications.

4. Discuss the implications of the model analysis results, emphasizing the key factors influencing the coordinated development of ecological well-being and providing insights for policymakers and practitioners.

4.1 Discussion Suggestions:

1. Provide a more detailed discussion of the temporal evolution analysis, interpreting the findings in the context of relevant theories and previous research. Discuss the implications of the observed trends and variations in the coupling coordination degree.

2. Discuss the spatial distribution patterns and regional disparities observed in the study, exploring the underlying factors and implications for policy interventions.

3. Highlight the key findings and their significance in advancing the understanding of the mechanisms driving the coordinated development of ecological well-being through industrial transformation.

4.2 Conclusions Suggestions:

1. Clearly summarize the main findings of the study, emphasizing the key factors and mechanisms influencing the coordinated development of ecological well-being in rural counties.

2. Provide a concise statement of the study's contributions and significance, highlighting the implications for sustainable rural development and ecological protection.

3. Offer suggestions for future research directions or areas that warrant further investigation, based on the limitations and knowledge gaps identified in the current study.

4.3 Policy Recommendations Suggestions:

1. Provide specific and actionable policy recommendations based on the research findings, addressing the challenges and opportunities identified in the study.

2. Justify each policy recommendation by linking it to the specific research results and the potential impacts on promoting the coordinated development of ecological well-being.

3. Discuss the practical implications and potential implementation strategies for the proposed policy recommendations, taking into account the local context and stakeholder involvement

Reviewer #2: A dual-fixed-space Durbin model is constructed to analyze the influence mechanism and spatial interaction effects of industrial transformation and upgrading on the coordination of ecology and well-being. The paper discussed the mechanism of promoting coordinated development of ecological wellbeing in rural counties through industrial transformation.

The structure of the article is clear and the process is explained well here. However, it could be improved in some detail.

1.The abstract can be further refined. For example, the research results in the abstract can be summarized into three points.

2.The contributions of this paper require further clarification and improvement. It is suggested to further improve the innovation points.

3.Some of the statements are written without reference; try to add reference with every statement in your paper.

4. In the discussion part, the author should further explain the reasons for these results.

5. The overall quality of English is good, but need to be checked carefully again. I suggest the authors should look for an English native speaker to further check the language of the paper.

6. Some fresh paper can be used as ref. eg: Liu, F., Sim, J. Y., Sun, H., Edziah, B. K., Adom, P. K., & Song, S. (2023). Assessing the role of economic globalization on energy efficiency: Evidence from a global perspective. China Economic Review, 77, 101897. https://doi.org/10.1016/j.chieco.2022.101897.

7.why nothing in the 3rd part of 2.3 Data sources?

6. PLOS authors have the option to publish the peer review history of their article (what does this mean?). If published, this will include your full peer review and any attached files.

Reviewer #1: No

Reviewer #2: No

---

## [Author Response · Author response to Decision Letter 0]

18 Aug 2023

Response to Editor and Reviewers

To Reviewer 1:

Thank you very much for your review and suggestions on the article; they have been greatly beneficial to the paper. We have made systematic revisions based on the changes you provided, and we have marked the modified areas in red. The specific changes are as follows:

1.The abstract can be further refined. For example, the research results in the abstract can be summarized into three points.

We have revised the abstract, taking your suggestions into account, condensing and categorizing the conclusion into 3 points, and restructuring the abstract to include the research objectives, research methods, research content, and main conclusions. The revised content can be found on the first page of the article. Once again, thank you for your suggestions for revision. They have been very helpful to us.

2.The contributions of this paper require further clarification and improvement. It is suggested to further improve the innovation points.

Thank you for your suggestions. We found that our previous manuscript might not have adequately described the core innovative aspects of this paper, so we have made corrections to the sections that describe the innovations. We have emphasized the innovation in the methods, perspectives, and content of the research, and highlighted the necessity and high value of this study. The specific modifications can be found on the page 2 of the article. Once again, thank you for your suggestions for revising the article.

3.Some of the statements are written without reference; try to add reference with every statement in your paper.

Thank you for your comments. Indeed, we discovered that some sentences did not have appropriate references to support them, so we have added citations where we felt they were needed in the article. The specific additions can be found on page 1, page 3, and page 4. If there are any areas where you still feel something is missing, we would be happy to make further additions.

4. In the discussion part, the author should further explain the reasons for these results.

Thank you for your suggestion. In the original manuscript, we only described the phenomenon without delving into its possible underlying reasons. Following your advice, we have revised this section of the manuscript. By considering both the phenomenon and its historical context, we conducted a deeper analysis of the essence of the phenomenon. The results of this analysis can be found on Page 17.

5. The overall quality of English is good, but need to be checked carefully again. I suggest the authors should look for an English native speaker to further check the language of the paper.

Thank you for affirming the level of English in our manuscript. We indeed spent a considerable amount of time polishing the language. We have once again sought out a scholar whose native language is English to make revisions to the manuscript's English. If there are still places that are not well-written, we would be very pleased to make further revisions.

6. Some fresh paper can be used as ref. eg: Liu, F., Sim, J. Y., Sun, H., Edziah, B. K., Adom, P. K., & Song, S. (2023). Assessing the role of economic globalization on energy efficiency: Evidence from a global perspective. China Economic Review, 77, 101897. https://doi.org/10.1016/j.chieco.2022.101897.

Thank you for your suggestion. We have updated the references in our manuscript. After consulting the literature you mentioned, we believe that it has a significant relationship with our research. The citation can be found on page 1. If you think there are other more suitable and recent references for this paper, we would be very pleased to make further updates.

7.why nothing in the 3rd part of 2.3 Data sources?

Yes, we made a mistake, and this section was erroneously deleted by us. We have corrected it and added back the complete information. Thank you for bringing this to our attention.

Thank you once again for taking the time to review and offer suggestions for our manuscript. Your valuable advice is crucial for us to enhance the quality of the article. We look forward to your next response. Please accept our most sincere blessings.

To Reviewer 2:

Thank you very much for your review and suggestions on the article; they have been greatly beneficial to the paper. We have made systematic revisions based on the changes you provided, and we have marked the modified areas in red. The specific changes are as follows:

About Abstract:

We have revised the abstract, taking your suggestions into account, condensing and categorizing the conclusion into 3 points, and restructuring the abstract to include the research objectives, research methods, research content, and main conclusions. The revised content can be found on the first page of the article. Once again, thank you for your suggestions for revision. They have been very helpful to us.

About Introduction

Thank you for your suggestions. We have identified many of the issues you pointed out in our introduction. Based on your recommendations for the manuscript, we have made the following modifications to the introduction:

(1) We have added some of the latest literature to make the logic of the article more coherent.

(2) We have rewritten the innovative points of the article, clarifying the objectives and advantages of the article, and emphasizing the importance of the research.

(3) We have revised the literature review to enrich this manuscript.

(4) We have made further language corrections to certain paragraphs, enhancing the readability of the article.

About Variable Selection, Model Setting, and Research Methods Suggestions:

Thank you for your suggestions. We have made certain revisions to this module:

(1) In addition to the original information about the data sources, we have included explanations for the accessibility of the data.

(2) We have provided detailed descriptions of the data processing methods and synthesis techniques.

(3) We have systematically explained the theoretical basis for the coupling coordination degree in the article.

(4) We have given a detailed overview of the application of the SDM model in the article.

About Regional Overview Suggestions:

Thank you for your suggestions. We have made certain revisions to this module:

We have provided a more detailed description and rewrite of the study area overview, taking into consideration not only the geographical features of the research area but also its economic conditions. We have emphasized the necessity of research in this region and the scarcity of county-level studies.

About Research Methods Suggestions:

Thank you for your suggestions. We have made certain revisions to this module:

(1) We have systematically described the spatiotemporal transformation trends of the coupling coordination degree.

(2) We have created a county-level transition matrix analysis diagram for the coupling coordination degree, exploring the transition status of the coordination.

(3) We have analyzed the reasons for the transition.

(4) We optimized the expression of the empirical conclusions and rewrote the impact mechanism for this module.

About Discussion Suggestions:

Thank you for your suggestion. In the original manuscript, we only described the phenomenon without delving into its possible underlying reasons. Following your advice, we have revised this section of the manuscript. By considering both the phenomenon and its historical context, we conducted a deeper analysis of the essence of the phenomenon. The results of this analysis can be found on Page 17.

About Conclusions and Policy Recommendations Suggestions

Thank you for your suggestions. We have made certain revisions to this module:

We have further revised the conclusion of the article, emphasizing the mechanism of impact, and we have also made corrections to the policy recommendations. Based on the model regression results, we have provided more appropriate policy suggestions. We have proposed policy recommendations to drive regional development through regional development and have provided suitable data support to ensure that the recommendations are trustworthy.

Thank you once again for taking the time to review and offer suggestions for our manuscript. Your valuable advice is crucial for us to enhance the quality of the article. We look forward to your next response. Please accept our most sincere blessings.

---

## [Decision Letter · Decision Letter 1]

24 Aug 2023

Research on the mechanism of promoting coordinated development of ecological well-being in rural counties through industrial transformation

PONE-D-23-20262R1

Dear Dr. Yang,

We’re pleased to inform you that your manuscript has been judged scientifically suitable for publication and will be formally accepted for publication once it meets all outstanding technical requirements.

Kind regards,

Fuyou Guo, (Ph.D.

Academic Editor

PLOS ONE

Additional Editor Comments (optional):

Reviewer 1 Comments :

Now the points are already addressed, the structure is ok, and the analysis is strong enough, so I suggest to accept it.

Reviewer 2 Comments :

I appreciate for the authors' revision. The existing paper has original content and worthy for publication in the journal. I can recommend it for a possible publication.

Reviewers' comments:

Reviewer's Responses to Questions

**Comments to the Author**

1. If the authors have adequately addressed your comments raised in a previous round of review and you feel that this manuscript is now acceptable for publication, you may indicate that here to bypass the “Comments to the Author” section, enter your conflict of interest statement in the “Confidential to Editor” section, and submit your "Accept" recommendation.

Reviewer #1: All comments have been addressed

Reviewer #2: All comments have been addressed

2. Is the manuscript technically sound, and do the data support the conclusions?

Reviewer #1: Yes

Reviewer #2: Yes

3. Has the statistical analysis been performed appropriately and rigorously? 

Reviewer #1: Yes

Reviewer #2: Yes

4. Have the authors made all data underlying the findings in their manuscript fully available?

Reviewer #1: Yes

Reviewer #2: Yes

5. Is the manuscript presented in an intelligible fashion and written in standard English?

Reviewer #1: Yes

Reviewer #2: Yes

6. Review Comments to the Author

Reviewer #1: I appreciate for the authors' revision. The existing paper has original content and worthy for publication in the journal. I can recommend it for a possible publication.

Reviewer #2: Now the points are already addressed, the structure is ok, and the analysis is strong enough, so I suggest to accept it.

7. PLOS authors have the option to publish the peer review history of their article (what does this mean?). If published, this will include your full peer review and any attached files.

Reviewer #1: No

Reviewer #2: No

---

## [Editor Report · Acceptance letter]

31 Aug 2023

PONE-D-23-20262R1 

Research on the mechanism of promoting coordinated development of ecological well-being in rural counties through industrial transformation 

Dear Dr. Yang:

I'm pleased to inform you that your manuscript has been deemed suitable for publication in PLOS ONE. Congratulations! Your manuscript is now with our production department. 

Kind regards, 

on behalf of

Associate professor Fuyou Guo 

Academic Editor

PLOS ONE